# Identifying a developmental transition in honey bees using gene expression data

**Bryan C. Daniels** [1] *, **Ying Wang**[2], **Robert E. Page Jr**[3,4], **Gro V. Amdam**[3,5]

**1** School of Complex Adaptive Systems, Arizona State University, Tempe, Arizona, United States of America, **2** Banner Health Corporation, Phoenix, Arizona, United States of America, **3** School of Life Sciences, Arizona State University, Tempe, Arizona, United States of America, **4** Department of Entomology and Nematology, University of California Davis, Davis, California, United States of America, **5** Department of Ecology and Natural Resource Management, Norwegian University of Life Sciences, Aas, Norway

* bryan.daniels.1@asu.edu

**Data Availability Statement:** All code needed to reproduce the analysis on our dataset and apply it to other datasets is available as a GitHub repository at https://github.com/Collective-Logic-Lab/landau. The experimental data are publicly available as a

## Abstract

In many organisms, interactions among genes lead to multiple functional states, and changes to interactions can lead to transitions into new states. These transitions can be related to bifurcations (or critical points) in dynamical systems theory. Characterizing these collective transitions is a major challenge for systems biology. Here, we develop a statistical method for identifying bistability near a continuous transition directly from high-dimensional gene expression data. We apply the method to data from honey bees, where a known developmental transition occurs between bees performing tasks in the nest and leaving the nest to forage. Our method, which makes use of the expected shape of the distribution of gene expression levels near a transition, successfully identifies the emergence of bistability and links it to genes that are known to be involved in the behavioral transition. This proof of concept demonstrates that going beyond correlative analysis to infer the shape of gene expression distributions might be used more generally to identify collective transitions from gene expression data.

## Author summary

A complicated biochemical choreography is responsible for both the behavior of cells that make up organisms and, at a larger scale, the behavior of cooperating groups like honey bee colonies. Individuals in these systems often take on specialized roles: cells assume specific types and bees perform specific tasks. Distinct roles are thought to emerge when differences among individuals are amplified. This amplification can create a transition from continuous variation to a case with two separate types. Identifying how and when these transitions occur has been challenging. Here, we create a method inspired by statistical physics to detect this type of collective transition using gene expression data. We demonstrate it by locating a known transition in honey bees between those that stay in the nest and those that leave to find food. We also show that the method can handle the type of large datasets that are now routinely generated in many areas of systems biology.

figshare data archive at https://doi.org/10.6084/m9.figshare.22696312.

**Funding:** This research was supported by the Research Council of Norway [335244] to GVA and NSF-DMS [1313312 and 1558127] to REP. BCD was supported by a fellowship at the Wissenschaftskolleg zu Berlin and by the ASU-SFI Center for Biosocial Complex Systems. The funders had no role in study design, data collection and analysis, decision to publish, or preparation of the manuscript.

# 1 Introduction

Social insects represent well-known examples of adaptive collective systems, combining the efforts of many individual actors to produce robust and adaptive aggregate behavior [1]. The allocation of tasks to individuals and, in some cases, specialization of individuals to particular tasks, often displays a sophisticated organization that promotes collective success [2]. This distributed coordination of effort is the result of a complicated process reaching from the level of gene regulation to social interactions. Mapping out the drivers of this process is a major challenge in understanding how collective behavior is successfully regulated in social insects, and connects to open questions more generally across living systems [3, 4].

In honey bees, specialization occurs typically among the worker daughters of a single queen. The largest distinction among workers is between those that mostly perform tasks within the nest (brood care, food processing, nest construction and maintenance, and colony defense) and those that leave the nest to forage. Bees typically begin their lives performing more in-nest tasks (nest bees), and in about their second to fourth week of life a major transition occurs in which they switch to foraging outside the nest (foragers) [5, 6]. Individual bees vary in the age at which they make this nest bee to forager transition, and while the transition can be reversed in individual bees if the colony's needs change, few normally do. This temporal bifurcation leads to an age-related division of labor [7].

The mechanisms behind this behavioral transition include interactions between genes and other factors expressed within each bee, interactions between bees, and interactions between bees and their local environment [5, 8–14]. At the individual level, the gene product Vitellogenin (VG, formerly known as an egg-yolk precursor) and the endocrine factor juvenile hormone (JH), and the interactions between the two, have been well studied [5, 12, 15]. The interaction takes the shape of a mutually suppressive feedback control system that appears to be at the core of the bifurcation of behavior between nest activities and foraging. Increasing levels of VG suppress JH in early life, while lower levels of VG in later life leads to higher titers of JH that further suppresses VG by inhibiting vg gene expression [5]. The behavior of this positive feedback control is consistent with the phenomenology of the transition, with either increases to JH (e.g. resulting from fewer interactions with other foraging bees as well as exposure to a variety of stressors) or decreases to VG (e.g. resulting from reduced food intake or excessive consumption of physiological nutrient stores during nursing tasks) leading to reinforcing feedback and the initiation of foraging. This general picture has recently been refined by situating VG and JH within larger gene networks involved in target of rapamycin and insulin signaling, ecdysone response, ovarian and neural activation [13]. Still, little is known about the full spectrum of genes and network dynamics that drive the transition from nest bee to forager and at least temporarily "lock" worker honey bees into the different roles.

An analogous phenomenon occurs in cell differentiation in multicellular organisms. Here, genetically identical cells perform distinct roles in the larger organism. Distinct cell types are "locked in" by regulatory interactions typically understood at the level of gene transcription. The dynamics of these interactions are thought to produce separate attractor states with different gene expression patterns that correspond to distinct cell types [16, 17].

In the language of dynamical systems, transitions among separate attractors are understood as arising from bifurcations that change the number of attractor states or from noise-induced hopping among co-existing attractors. In the cellular differentiation literature, statistical tools have been developed for use with gene expression data to identify transitions that create new attractor states [17–20]. These studies have shown that multiple cases of cell state change are consistent with dynamics that pass through or near a continuous bifurcation, also known as a critical transition. This type of bifurcation is known to result from positive feedback loops in

many biochemical networks [21, 22]. Such a bifurcation can be identified by monitoring for increased correlated fluctuations in gene expression before the transition [17, 18] or by testing for evidence of bimodality in gene expression after the transition [19, 20]. More generally, such critical transitions are theorized to be ubiquitous across biology as mechanisms for controlling collective behavior [3, 4, 23–26].

In honey bees, studies linking gene expression to behavior have tended to focus on searching for individual genes whose expression correlates with behavior [14, 27, 28]. This contrasts with a collective-scale view of multiple interacting genes across multiple individuals that we anticipate will be necessary to find evidence of critical transitions.

In this study, we analyze gene expression data across an ensemble of individual honey bees and across a series of developmental timepoints to examine genetic-scale indicators of the transition from in-nest tasks to foraging. We develop a method for identifying critical transitions based on the Landau theory of phase transitions in statistical physics and the theory of bifurcations in dynamical systems. At each timepoint, using measurements of gene expression across multiple bees, we are able not only to estimate mean expression values and their co-variance, but to infer the shape of the distribution of expression values. This shape can be expected to begin as unimodal as bees initially emerge from pupation, with gene expression values fluctuating around a common mean. As the adult bees develop, increased variation in behavior could be connected to continuous variation in gene expression, or at some point expression values could split into two well-defined groups (e.g. nest bees and foragers) corresponding to discontinuous variation. Such a transition would be indicated by a change from unimodality to bimodality in the distribution of expression values, which our method explicitly identifies. Our method is designed to work with high-dimension-low-sample-size data as is common in transcriptomic data; it does not require prior knowledge of important genes; and it is formulated using a Bayesian interpretation to provide precise statistical meaning with respect to model selection and transition identification. We use our method to answer the questions of whether and when the bees' transition to foraging is visible in gene expression data, whether it is consistent with a continuous bifurcation transition, and which genes are most associated with such a transition.

## 2 Results

We examined gene expression profiles of $N_{\text{samples}}$ = 16 bees at 5 developmental timepoints, from 1 to 15 days old, spanning the transition in behavioral development from all bees remaining in the nest to some bees leaving the nest to forage. To critically test the concepts of our statistical model, we focused on a set of genes $N_{\text{genes}}$ = 91 that we expected to be involved in distinguishing the nest bee and forager behavioral phenotypes through their actions on and with VG, a known determinant of foraging onset. Some of these genes have known correlations with VG production and with each other, while others were suspected of being involved through their action in common gene networks. Additional genes were selected because they are located within the genetic map confidence limits of known quantitative trait loci (QTL) shown to have effects on ovaries, VG production, and foraging behavior [29], or involved in stress or immune-associated pathways that have been correlated with the behavioral transition before [30]. We predicted that the expression of sets of genes affecting foraging should diverge from their expression in nest bees leaving a detectable bimodal distribution of gene expression representing the bifurcation.

We tested for bimodality consistent with a continuous transition using a Bayesian analysis. In particular, we compared the goodness-of-fit of a unimodal Gaussian distribution with the form of bimodal distribution expected near a continuous transition (see Methods). We were

able to perform this test with low sample sizes ($N_{\text{samples}} < N_{\text{genes}}$) because bistability is expected to begin along the dimension with the largest variance (the first principal component; see Methods section "Dynamical model of gene interactions"), and we could thus focus on identifying bimodality along this single dimension.

We first demonstrate proof of the principle that this method can successfully infer a transition to bistability in simulated data from a simple model of noisy gene regulatory dynamics with $N_{\text{genes}} = 91$. Fig 1 displays the fraction of simulations in which evidence of a transition was successfully identified as a function of the strength of interactions that induce a transition. While a larger number of samples can identify the transition earlier ($N_{\text{samples}} = 100$; red points), even with the smaller number of samples that we have here ($N_{\text{samples}} = 16$; orange points), we expect to be able to identify such a transition once the bimodality is more pronounced.

Applying this analysis to the honey bee expression data, we find a transition into two distinct groups that is statistically visible at age 10 days and more apparent at age 15 days (Fig 2). Specifically, a Bayesian Information Criterion comparing the unimodal Gaussian distribution to the bimodal distribution expected near a transition strongly favors bimodality at these later ages (Fig 2B; see Methods for more details). In Fig 3, we plot for these two ages the gene expression data and inferred transition distribution along the bistable dimension (corresponding to the first principal component; see Methods). The data and the inferred distribution in

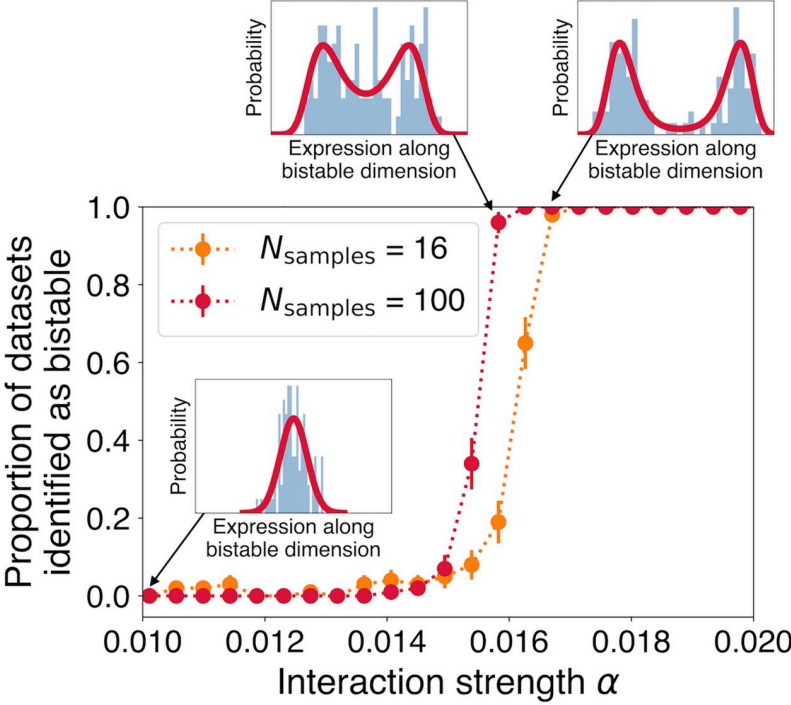

**Fig 1. The method successfully identifies the transition state in simulated data.** As interaction strength $\alpha$ increases in a simple dynamical model of gene regulation (see Eq (19) in the Methods), gene expression levels projected along the first principal component transition from a unimodal to a bimodal distribution (three insets show simulated data as blue histograms and the best-fit distributions as red curves). A statistical comparison between a Gaussian distribution and the distribution shape expected near a continuous transition reliably identifies the transition state once the bimodality is sufficiently pronounced (red and orange points). Increasing the number of data samples allows identifying the transition state when bimodality is less pronounced (compare red and orange points). Error bars indicate estimated standard deviation of the mean given 100 simulations.

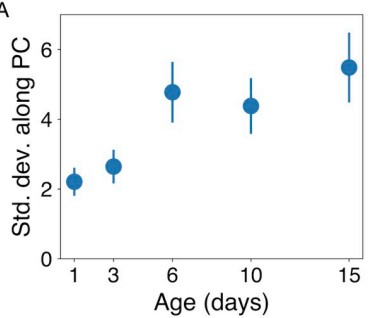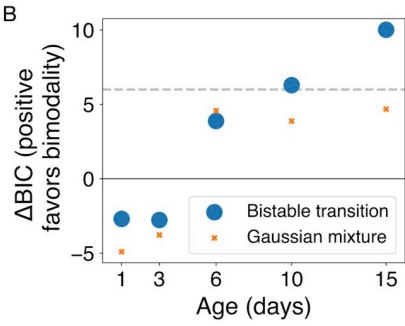

**Fig 2. In honey bees, both the variance of gene expression and the strength of evidence for bistability grow over developmental time.** A: The standard deviation of gene expression along the principal component increases during development. Error bars show standard errors ($N_{samples}$ = 16). B: The Bayesian Information Criterion measure ΔBIC quantifies the strength of evidence in favor of the bistable transition distribution as compared to a unimodal Gaussian, with positive values favoring bistability (blue circles; see Methods Eq (16)). We interpret ΔBIC values larger than 6 (horizontal dashed line) as strong evidence for bistability. We also compute ΔBIC that compares a Gaussian mixture ($n$ = 2) with the unimodal Gaussian, which identifies weaker evidence for bimodality (orange *X*s; see section 3).

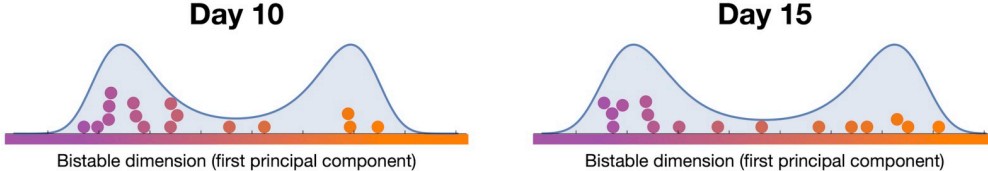

**Fig 3. Bistability along the first principal component.** Gene expression data from 16 bees at age 10 days and 15 days projected along the first principal component (colored circles; log-transformed data). Colors correspond to distance along this dimension, with orange chosen to represent the low *vg* state indicative of foragers. The fit Landau distribution is shown in blue.

this dimension are characterized by bees that fall roughly into two clusters, with a few bees that are transitional between the two clusters.

Different genes contribute differentially to defining the bistability. Sorting by the fraction *s* of each gene's variance that lies along the bistable dimension highlights those genes whose expression values give the most information about the side of the bistability on which each bee lies (Fig 4 and Table 1). In Fig 4, we also display the measured expression levels of the most informative genes, with datapoints from each bee colored according to their positions along the bistable dimension. Comparison with known gene expression patterns suggests that purple points correspond to bees predisposed to in-nest behavior, and orange points to those predisposed to foraging behavior [5, 31].

## 3 Comparison to other potential methods for identifying a transition

A family of related methods for detecting bimodality would correspond to fitting bimodal distributions of different shapes along the first principal component. One simple choice would be a mixture model that combines two Gaussian distributions. We compare our derived "Landau" distribution to this Gaussian mixture distribution (fitting the location of each Gaussian, a common width, and their relative weight, producing 2 extra degrees of freedom compared to the single Gaussian model). In Fig 5, we demonstrate using simulated data that this

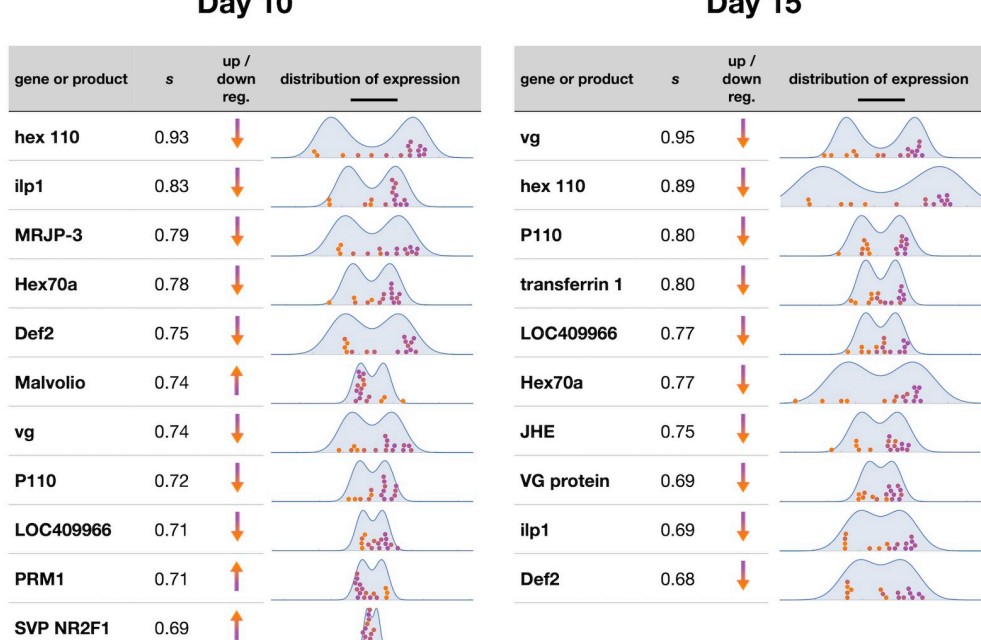

**Fig 4. Individual genes associated with the bistability.** A handful of individual genes or gene products have a large proportion $s$ of their variance along the bistable dimension. Here we highlight all genes with $s > 2/3$. Orange and purple colors correspond to bees on two sides of the bistability, with orange chosen to represent the low *vg* state that is indicative of foragers. In the column "up/down reg.", we indicate whether the gene is up or down regulated in the orange state as compared to the purple state. Expression data for these genes are shown with the same colors for individual bees as in Fig 3, along with the marginalized fit distribution in blue (log-transformed data, with scale bar corresponding to an expression ratio of 10).

distribution performs significantly worse in detecting the transition to bimodality and in fitting the distribution of gene expression values. Comparing the best fit of the two distributions to 50,000 samples from the model close to the transition (Fig 5B), the Landau distribution is a notably better approximation to the sampled data than the Gaussian mixture model. Even for this large number of samples, the Landau distribution is not statistically distinguishable from the sampled data (KS statistic = $4.2 \times 10^{-3}$, $p = 0.33$), while the Gaussian mixture distribution, with one more parameter than the Landau model, is significantly different (KS statistic = 0.017, $p < 10^{-10}$). Furthermore, the orange points in Fig 2B show that the Gaussian mixture model is not as statistically favored as the Landau model in fitting our experimental data.

Another related technique, explored in the context of cell differentiation in Ref. [17], looks for signatures of a continuous transition *before* the split to bimodality, related to increased variance in the expression of particular genes. The method is simpler than ours in that it only requires measuring pairwise correlations among genes and bees, and does not explicitly measure the shape of the distribution of expression values. The method has the advantage of explicitly incorporating the increase in variance one expects to find when an existing attractor is destabilized, which can help to distinguish the case of attractor destabilization from the case of hopping between co-existing attractors. Applied to our data, a relative increase of the resulting "transition index" at days 10 and 15 (see Fig 6) is also suggestive of a transition state, but only when limited to genes that we identify as most informative through our measure $s$. We note that the original formulation of this measure did not assume that the most important genes were known *a priori* [17], though this seems to be necessary to identify the transition when

**Table 1. Genes whose variance in expression is most aligned with the transition dimension.** Here we list the top 20 genes, at age 10 and 15 days, ordered according to *s*, the fraction of the gene's variance that lies along the bistable dimension (see Methods Eq (17)).

| Day 10 | | Day 15 | |
|---|---|---|---|
| **gene** | **s** | **gene** | **s** |
| hex 110 | 0.93 | vg | 0.95 |
| ilp1 | 0.83 | hex 110 | 0.89 |
| MRJP-3 | 0.79 | P110 | 0.80 |
| Hex70a | 0.78 | transferrin 1 | 0.80 |
| Def2 | 0.75 | LOC409966 | 0.77 |
| Malvolio | 0.74 | Hex70a | 0.77 |
| vg | 0.74 | JHE | 0.75 |
| P110 | 0.72 | VG protein | 0.69 |
| LOC409966 | 0.71 | ilp1 | 0.69 |
| PRM1 | 0.71 | Def2 | 0.68 |
| SVP NR2F1 | 0.69 | PRM1 | 0.65 |
| AGO2 | 0.61 | TOR | 0.64 |
| Hymenoptaecin | 0.59 | Hymenoptaecin | 0.62 |
| SmG | 0.58 | Malvolio | 0.59 |
| TOR | 0.54 | TYR1 | 0.58 |
| OA1 or OAR | 0.54 | E74 | 0.45 |
| VG protein | 0.53 | Kr-h1 | 0.44 |
| USP (RXR) | 0.53 | cad | 0.42 |
| AKHR | 0.53 | InR1 | 0.39 |
| Kr-h1 | 0.52 | AGO2 | 0.35 |

applied to our data. Though our method is somewhat more computationally involved, it is able to identify individual genes that are related to transition states, infer the shape of the transition distribution, and produce a statistically interpretable measure (ΔBIC) of the likelihood of bistability given the data.

Finally, we compare with a method that attempts to identify transitions before they happen based on "state-transition-based local network entropy" (SNE) [18]. The method is based on

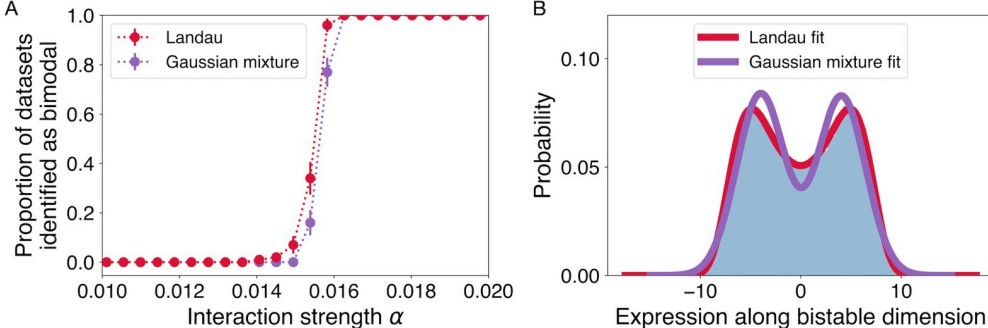

**Fig 5. Fitting to the Landau probability density function performs better than a mixture of two Gaussians.** In tests with simulated data, the Landau method (A) identifies the transition state sooner (at smaller $\mu$, when the bistable states are closer to one another; shown here fitting 100 samples), and (B) fits the transition distribution much more closely (shown here fitting 50,000 samples at $\mu = 0.0158$; simulated data in blue histogram compared to best fits of Landau distribution and Gaussian mixture distribution shown as solid curves). Error bars in (A) indicate estimated standard deviation of the mean given 100 simulations.

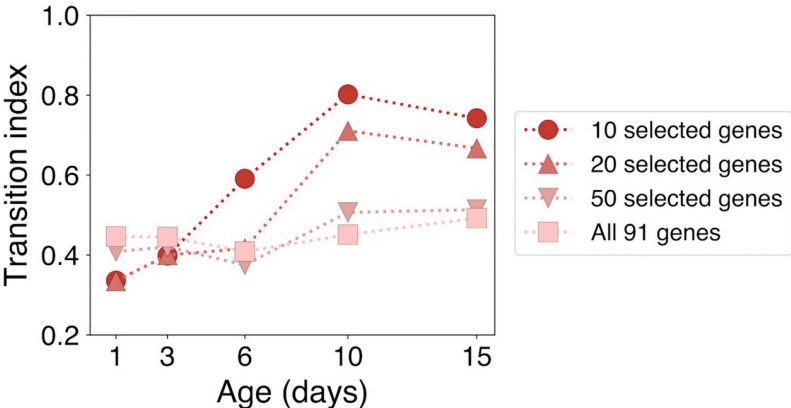

**Fig 6. Existing "transition index" measure also suggests a transition when focused on particular genes.** The transition index defined in Ref. [17] requires selecting a set of genes that are known *a priori* to be involved in a transition. Applying this measure to our data and restricting to genes identified by our method, the largest values occur at age 10 and 15 (red circles). This corroborates our method, which does not require selecting specific genes, yet locates a transition at a similar time. In contrast, when including data from all 91 genes (pink squares), the signal in the transition index is washed out.

identifying correlated fluctuations over time (looking for large changes in SNE within subsets of genes), and requires knowledge of the network of gene interactions to restrict the computation of network entropy to local gene subsets. We do not have such an interaction network in the honey bee case, so we instead compare our method by applying it to a dataset already analyzed by the SNE method [18]. These data track the transition in humans from healthy liver function to advanced stages of liver cancer (hepatocellular carcinoma, HCC) [32], with gene expression profiles including tens of thousands of genes from multiple tissue samples in healthy controls and in 7 progressive stages of disease. The SNE approach found evidence for a pre-critical state in the "very early HCC" stage [18]. While SNE looks for fluctuations indicative of a transition to bistability that is imminent but has not yet happened, our approach is designed to identify bistability as soon as possible after the transition has occurred. We therefore expect our method to identify bistability at the same stage or soon after the stage identified by SNE. When we apply our Landau approach to the data (Fig 7), we indeed find strong and growing evidence of bistability starting at the "early HCC" stage.

## 4 Discussion

We show that expression data from 91 genes, taken from 16 honey bee workers sampled at 5 different time intervals that span the first 15 days of life, demonstrate bimodality within time intervals that becomes more pronounced with age. We hypothesize that the observed bimodal distributions of gene expression correspond to known temporal dynamics of behavior. Days 10 to 15 correspond to a known developmental time point at which individual bees transition or prepare to transition from in-nest activities to foraging [6]. Days 10 and 15 demonstrate the strongest evidence for bimodality, suggesting that the two modes represent the transitioning of bees from those that engage in nest activities into those that forage.

We propose that the sets of genes defining these bimodal structures are working together in networks with bistability associated with the distinct behavioral states [5, 15, 33]. The gene networks have two stable states that are driven by the JH/VG positive feedback control system. The JH/VG feedback system drives gene expression that sculpts the physiological and

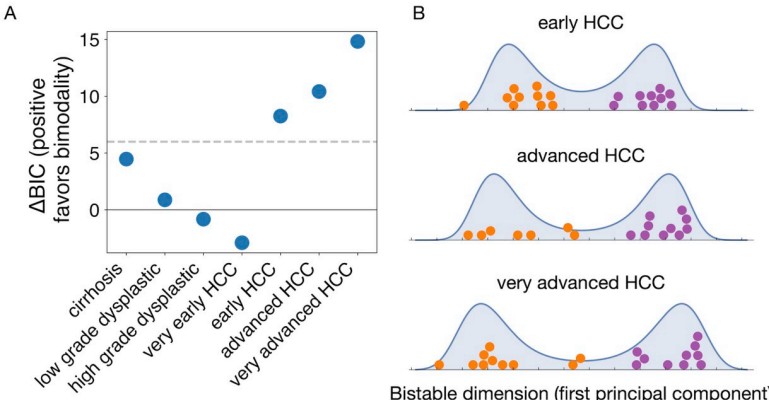

**Fig 7. Applying the Landau method to hepatocellular carcinoma data demonstrates its relationship with methods that rely on the increase of correlated fluctuations.** (A) Combining control samples from healthy livers with samples from various stages of liver cancer (data from Ref. [32]), we find increasingly strong evidence of bimodality starting at the "early HCC" stage. The SNE approach, designed to find evidence of transitions before they happen, identified critical fluctuations at the "very early HCC" stage [18]. (B) Along the bistable dimension, increasingly separate clusters of gene expression are visible. The ground-truth group membership of each sample is indicated by color: purple for control samples and orange for disease samples.

behavioral changes in worker behavior. The resulting switch is steep as a consequence of the feedback mechanisms affecting blood titers of VG and JH. During the nest stage, blood levels of VG initially increase but eventually decline due to dynamics including depletion (its use in brood food production) and/or suppression of synthesis due to reduced food intake/availability, disease or stress. As VG levels decline, JH titer increases, driving the production of VG even lower. This positive feedback is behind the steep switch and a route to the transition of the bee into the forager state. Note that VG may directly affect expression levels of other genes via DNA-binding capability [34]. There is also a JH and VG independent pathway to the forager state, but its initiator(s) is unknown [5, 35].

The transition is affected by the external environment through feeding activities and stressors: Feeding more larvae can result in more rapid VG depletion and an earlier onset of foraging. Workers in colonies that have stored insufficient pollen, the source of protein, initiate foraging earlier in life, presumably due to a deficiency in VG levels. Workers in colonies with fewer foragers accelerate the onset of foraging, presumably by having a reduction of some unknown inhibitor passed from foragers to younger bees [36] or by the lack of nutrients to support colony growth. Similarly, pathogens that reduce VG production by, for instance, reducing nutrient uptake over the gut or consumption of resources in immune activation lead to earlier foraging onset [37].

Given this context, our list of genes that align most strongly with with the gene network bistability (Fig 4 and Table 1) is one of "usual suspects." Most of the genes listed have known involvement in lipid and lipoprotein synthesis and metabolism, are lipoproteins found in the hemolymph or hypopharyngeal glands of nest bees involved directly in larval and adult nutrition, regulate the JH/VG switch between nest bees and foragers, or are known "immunity genes" involved in responses to pathogenic organisms. This result is also an outcome of our initial selection of the 91 genes, which in essence focused on "usual suspects" to provide a strong basis for testing our statistical methodology.

More specifically, VG, major royal jelly protein 3 (MRJP-3), and the hexamerins (hex 100 and Hex70a) are lipoproteins that are metabolized prior to the onset of the foraging state. Elements of the insulin signaling pathway include insulin like peptide 1 (ilp1), its receptor (InR1),

and P110. Insulin-like signaling (IIIS) acting in the fat body has been shown to be involved in VG production and the transition from nest bee to forager. The target of rapamysin (TOR) pathway cross-talks with IIS signaling in the regulation of JH, while at least TOR is influencing VG levels more directly presumably by communicating nutritional context information to the VG production system [38], which is highly nutrient sensitive [39]. Transferrin 1, hymenop-taecin, argonaute 2 (AGO2), and defensin-2 (Def2) are genes associated with immunity. Their role may be one of adjusting to the change in pathogenic challenges that take place when a bee transitions from the protected nest environment to foraging. Malvolio is a manganese trans-porter that affects sucrose sensitivity, and through its effect on mn2+ in the brain presumably influences neuromodulators involved in the onset of foraging. Octopamine (OA) is involved in the onset of foraging and increases sensory motor responses involved in foraging; OAR is its receptor.

The pattern we reveal for juvenile hormone esterase (JHE), strongly correlated with bist-ability in bees of age 15 days, might be particularly novel and intriguing. JHE provides oppor-tunity to metabolize JH inside cells, and this action may block cellular responses to bursts in circulating (blood) JH titers. We show that JHE transcript levels decrease in the low vg state indicative of foragers, thereby making these individuals potentially more sensitive to JH. Such sensitivity would add to the positive feedback control system of VG and JH, which secures that transitioning individuals stay committed to their new behavioral role.

What causes this split into two groups? The correlation of these genes and the bistability can be explained in multiple ways. The transition could be controlled by an internal temporal program, or "physiological clock", running independently within individual bees [40]. Alter-natively, the bifurcation could arise solely via external positive feedback, with amplification of individual differences in foraging thresholds through interactions with the nest, the environ-ment, and other bees, similar to that seen in other examples of division of labor [9]. Perhaps most likely is a combination of these two mechanisms: internal dynamics that create a steep switch between nest bees and foragers, with the timing of this switch greatly influenced by the environment [5, 14]. The toy model of gene interactions that we use to test the statistical infer-ence is one way to represent dynamical mechanisms that can amplify differences in gene expression to create distinct types, which could be further amplified at the behavioral scale.

Our statistical method of Bayesian model selection for identifying bistability is designed to impose the least structure possible to explain the data: we start from the simplest possible prob-ability distribution (Gaussian) and compare this to the case in which we add the lowest order correction term consistent with a transition. This simple representation is useful when we have limited data (in this case, 16 bees per timepoint). In future cases with more data, more precise versions may be fruitful, including those that test whether expression data are consis-tent with a continuous or discontinuous transition.

Existing methods for identifying critical transitions from gene expression data have largely focused on the increase of correlated variability near a transition, and before the transition induces a split into two clusters [17, 18, 41]. In contrast, and similarly to Ref. [19], the method we develop here goes beyond pairwise correlation and looks for a characteristic shape of bimo-dality during and after this split. Our method remains statistically robust, working in the regime of "high dimension low sample size" [42] that is common in gene expression studies. In the hepatocellular carcinoma example (Fig 7), we show that the method is able to handle large datasets with tens of thousands of genes. The method has no free parameters, unlike many clustering methods, and it is interpretable in a Bayesian analysis to quantify the evidence favoring bistability.

An advantage to our approach is the ability to include a large number of genes that may or may not be related to a transition and then identify those individual genes that are most

informative about the transition. Previous approaches to identifying transition states from transcriptomic data have also attempted this, though they have some limitations. One approach that analyzes critical fluctuations of gene expression across time has been used to identify genes involved in a transition [18] but requires detailed knowledge of protein interaction networks. Second, the transition index approach that we compare to in Section 3 has been shown, when applied to other datasets, to be robust to including large numbers of genes that may or may not be closely related to a given transition [17]. Yet this does not appear to be the case for our data, in which the transition is only detectable by the transition index measure once less informative genes have been filtered out (Fig 6).

We expect that our approach could be useful for interpreting the dynamics of gene expression in other organisms and perhaps even more generally across systems biology. First, in the development of multicellular organisms, the divergence of progenitor cells into differentiated cells can also be well-represented using bifurcation theory [17, 19, 43, 44]. Current approaches for identifying cell types from gene expression data typically separate the data into distinct types using clustering methods (e.g. [19, 42, 45]). While a clustering approach is suitable for cases in which cell types are highly distinguishable, clustering will break down as cell types become less distinguishable near a continuous transition. Our method, in contrast, is designed to work near the transition point. This type of analysis could be used, for instance, to use gene expression data to more precisely define the time in development at which one cell type becomes two.

Other collective decisions, such as in neural systems, are thought to arise from continuous bifurcations as well [46], and our method could be used directly on activity data to identify the individual components (e.g. neurons) that may drive these transitions.

More generally, the method could be useful whenever there is variability among individual samples that may undergo a continuous transition into separate types, and it is particularly valuable for interpreting high-dimensional datasets. Any type of -omics data could be used for the analysis, given two basic requirements. First, there must be sufficient separation of timescales that an equilibrium approximation is sensible, with fast dynamics that are roughly equilibrated over the slower timescale of the transition to bistability. Second, the data must include a sufficient number of sampled individuals at each timepoint (a minimum of roughly 10) in order for the bistable Landau distribution to be statistically distinguishable from a simpler Gaussian distribution.

If our method fails to detect a transition, this indicates either that there is no transition to bistability or that there is a transition of a type that cannot be well-described by the Landau distribution. Further work will be necessary to distinguish possible cases that could evade detection in this way. As a start, we highlight three assumptions in our derivation (set out more precisely in sections 5.3 and 5.4) that may not be met by some biological systems and therefore may lead to transitions that would not be well-described by our method. First, we assume a symmetric transition, with no bias toward either of the bistable types. Such a bias could be better represented by a more complicated distribution, but would require more parameters and therefore more data to reliably infer. Second, our derivation assumes continuous time and continuous variables, whereas some regulatory systems (with, for instance, bursty dynamics or highly nonlinear saturating functions) may not easily conform to this. Finally, we assume that the interaction network is sufficiently dense for a Landau-type mean-field theory to be a good approximation. In spatially organized low-dimensional systems studied in statistical physics, correlated fluctuations are known to cause mean-field models to become invalid [47], such that the expansion in Eqs (5) and (6) is not justified. This could potentially lead to a transition with a different form than we derive. On the other hand, mean-field theory is known to work well even in relatively small spatial dimensions (e.g. 4 or larger for the well-

known Ising model), when connectivity becomes sufficiently dense. Due to the prevalence of feedback and relatively dense connectivity of biological networks, we are optimistic that mean-field theory represents a good approximation.

To summarize, in the context of animal behavior, our results add a new layer to understanding how the strategy of task specialization plays out in biological collectives. Even when individuals share the same genetic code and environment, their traits can diverge into separate tracks. Similarly to differentiation in multicellular organisms, this divergence is visible both at the scale of functional characteristics and at the scale of gene expression. Furthermore, the simplest model of critical bifurcation provides a good description of how this divergence occurs. The generality of this phenomenology suggests that such critical transitions may be a common mechanism within biology, making use of the emergent properties of strongly interacting dynamical networks to generate reproducible diversity. Further work is needed to detail how those networks are successfully constructed and regulated in an evolutionary context.

## 5 Materials and methods

### 5.1 Gene expression data

The bee collection was conducted during March 18th to April 7th, 2016. Five strong wild type colonies were chosen as sources of newly emerging bees. Two small nuclear colonies with 9 frames in each were used as recipient colonies that had similar supplies of pollen and honey, and similar numbers of brood and adult bees. The emerging frames from source colonies were put into incubator (37°C) and about 200 newly emerged bees were collected, marked in a color on the bee thoraxes and distributed evenly to the recipient colonies. This procedure was repeated for three consecutive days. About 600 newly emerged bees were introduced in the recipient colonies from March 19th to March 21st, 2016. Ten each of 1 day old, 3 day old, 5 day old, 10 day old and 15 day old bees from each recipient colony were collected in the following days. Therefore, there were total 20 sample bees for further experiment from two recipient colonies for each age. RNA was extracted from 16 randomly selected bees of each age using TRIZOL method. The quality of RNA was measured by Nanodrop and 260/280 was between 1.85–2.00, indicating good quality of RNA. RNA samples were sent to the University of Arizona Core lab for Nanostring analysis, a method to quantify the copy of RNA for the target genes.

94 genes (see S1 Spreadsheet file) were selected based on prior knowledge and the potential association between these genes, VG production, and social behavior. Genes were selected from the honey bee *Apis mellifera* assembly and gene annotation information at NCBI (https://www.ncbi.nlm.nih.gov). Genes were prioritized using a pragmatic approach similar to Refs. [48, 49]. In brief, our strategy integrated information on a specific biological process (the nurse-forager transition) with information on genes that were suggestively, correlatively, or potentially causally involved. Thus, gene selection relied on a process of extensive literature review as well as interdisciplinary biological expertise. The gene prioritization was reviewed by three members of the interdisciplinary team (YW, GVA, and REP) to minimize operator biases. Four housekeeping genes, actin (GB17681), rp49 (GB10903), GAPDH (LOC410122) and RPS18 (LOC552726) were used for sample normalization. Finally, the vitellogenin (VG) protein was measured in the honey bee hemolymph based on the method in Ref. [50].

Geometric means of the housekeeping genes were used to calculate normalization factors to minimize the noise from individual genes using nSolver software provided by the company. We use log-transformed values of the raw expression data for all analyses.

## 5.2 Conceptual overview of statistical method

The expression of gene products across individual bees is correlated, with correlations arising from a complex set of dynamics. These dynamics include both direct interactions among measured genes and common factors of influence external to the measured genes. Changes to these dynamics (e.g. caused by stronger interactions among genes during certain phases of development, or shared interactions with a common environment) can lead to increased variance in gene expression. This additional variance is also correlated, forming patterns consisting of sets of genes with expression that moves in tandem. In the case of a transition into two distinct types, we expect that this correlated variance will additionally display bimodality. That is, two separate patterns of gene expression will coexist, with few individuals having gene expression levels that are a mixture between the two types.

In the following description of methods, we first build a simple model of gene expression dynamics that displays this characteristic transition to bimodality. We demonstrate that the separate patterns of gene expression corresponding to this bimodality are given by the dimension of largest variance (first principal component), as can be extracted from gene expression data by principal components analysis. We then show that the distribution of gene expression values along the principal component has a specific shape near the transition, which we derive. By fitting the observed data to this distribution, we determine whether gene expression across a population of individuals of the same age displays evidence of a bistable transition.

In our simple dynamical model, we think of interactions as capturing dependencies between gene expression within an individual bee. Neglecting interactions between bees simplifies the interpretation of the resulting probability distributions, as each sampled bee represents an independent sample. Yet interactions between bees are known to bias the transition. The simplest interpretation we could hope for is that these interactions are small, so that the distribution we observe experimentally is a good approximation for the case of non-interacting bees. This interpretation more easily explains the fact that bees transition from in-hive to foraging activities even when they are isolated from others. In the most difficult case, interactions between bees could be mostly responsible for the feedback that leads to bistability, with corresponding changes to gene expression within each bee that are driven by but do not cause the transition. Even in this case, our method provides a way to characterize the transition phenomenologically in terms of the degree of bistability and the genes that are associated with it.

## 5.3 Dynamical model of gene interactions

Near a transition from a system with continuous variation into one displaying distinct types, one expects the emergence of a bimodal distribution with a particular shape. In this section, we derive to lowest order the form of this distribution for the case of the dynamics of a densely connected gene regulatory network.

We start with a general form of dynamics for the (log-transformed) concentrations of the measured gene products, arranged in the $N_{\text{genes}}$-dimensional vector $\vec{x}$:

$$\frac{d\vec{x}}{dt} = F(\vec{x}, \vec{y}, t), \tag{1}$$

where $\vec{y}$ consists of other unmeasured variables and $t$ is time. We assume a separation of timescales such that, at each developmental timepoint, we are measuring an equilibrium distribution, which slowly changes as a function of developmental time. This means $F$ depends on the long developmental time $T$ (on the timescale of days), but the distribution of $\vec{x}$ at each $t$ will

not depend on $t$ defining gene transcription dynamics (on the timescale of seconds):

$$\frac{d\vec{x}}{dt} = F_T(\vec{x}, \vec{y}). \qquad (2)$$

The $\vec{x}$ we observe will then be restricted to be near attractors of $F_T$, determined by taking $t \to \infty$. Without the ability to measure the dynamics of more complicated attractors on this fast timescale (limit cycles, strange attractors, etc.), we will focus our attention on stable fixed point attractors $\vec{x}^*$, where

$$F_T(\vec{x}^*, \vec{y}^*) = 0. \qquad (3)$$

If we were to observe only these fixed point attractors, and for fixed hidden variables $\vec{y}^*$, we would erase all information about hidden variables and about dynamics at the fast time-scale. This corresponds to a description of the system we could get if we measured one bee per developmental timepoint.

We now want to incorporate information about variance in the observed $\vec{x}$ coming both from the effects of hidden variables $\vec{y}$ and from other fast-timescale dynamics. When these effects are small and temporally uncorrelated, we can approximate them as adding uncorre-lated noise to $F_T$:

$$\frac{d\vec{x}}{dt} = F_T(\vec{x}) + \vec{\xi}, \qquad (4)$$

where $\vec{\xi}$ is chosen from a multidimensional Gaussian with some fixed covariance. Assuming this variance is sufficiently small (compared to nonlinearities in $F_T$), the system stays near the fixed point $\vec{x}^*$. In this case, dynamics can be approximated using a Taylor expansion of $F_T$ around $\vec{x}^*$ (related to the Hartman–Grobman theorem [51])

$$F_T(\vec{x}) = \Lambda\, \delta\vec{x} + O(|\delta\vec{x}|^2), \qquad (5)$$

where $\delta\vec{x} = \vec{x} - \vec{x}^*$. The assumption that we are at a stable fixed point is equivalent to the statement that the eigenvalues of the matrix $\Lambda$ are all negative. The solution of these dynamics is a multidimensional Gaussian. This corresponds to the analysis performed in Ref. [17], where the existence of a critical point corresponds to an eigenvalue of $\Lambda$ becoming close to 0, such that there is a measurable large variance in $\delta\vec{x}$ along the corresponding eigenvector.

We take the next step in the expansion by looking at what happens when all of the above assumptions hold but we additionally want to treat the super-critical case, when an eigenvalue of $\Lambda$ becomes larger than 0. Our goal is to find the form of the simplest possible symmetry breaking transition that leads to two distinct modes of gene expression. Still assuming the noise is small means $\vec{x}$ stays near the fixed point except along the eigenvector $\hat{v}$ with eigenvalue greater than 0. Along that direction, the variance is largest, so this dimension will correspond to the first principal component. Further, because the dynamics are locally unstable to lowest order, the behavior along $\hat{v}$ is no longer determined only by the lowest-order expansion given in Eq (5). It will instead be determined by the lowest-order term that cures the divergence of the variance in that direction. (There are other ways the transition could happen such that this divergence is not cured (or not cured in a way that keeps $\delta\vec{x}$ relatively small). This corresponds, for instance, to a saddle-node bifurcation, in which the transition would cause the system to suddenly change to some other distant attractor. That is, there are transitions that would not be identified by our method. Our method is designed to detect only the simplest symmetric, continuous transitions.) Next will generally be a term of order $|\delta\vec{x}|^2$, but this still allows a diver-gence of $\delta\vec{x}$ along either positive or negative $\hat{v}$. In addition, we restrict our analysis to retain

the symmetry of $F_T(\delta\vec{x}) = F_T(-\delta\vec{x})$, so that the system will be equally likely to move in the positive or negative $\hat{v}$ direction at the transition, corresponding to a symmetry breaking that is unbiased. It is also possible to include a bias term that leads to one of the final attractors being favored over the other [19], but we omit that possibility here to produce the simplest possible symmetry breaking transition. For these reasons, we assume the second-order term is zero. The lowest-order term that does cure the divergence is of order $|\delta\vec{x}|^3$; we write

$$F_T(\vec{x}) = \Lambda\,\delta\vec{x} + \Gamma(\delta\vec{x}\cdot\hat{v})^3\hat{v} + O(|\delta\vec{x}\cdot\hat{v}|^4), \tag{6}$$

with $\Gamma < 0$. This corresponds to a so-called "supercritical pitchfork" bifurcation in dynamical systems theory, where one stable fixed point changes continuously into three fixed points, two stable separated by one unstable [52, 53]. The resulting two stable fixed points represent two distinct types of gene expression—phenotypes or "cell fates" in the cell differentiation literature—into which the single fixed point in gene expression splits. Finally, we note that even higher-order terms would be necessary if $\Gamma > 0$, which corresponds to a subcritical pitchfork [43], but we again omit that possibility to produce the simplest possible symmetry breaking transition.

## 5.4 Form of Landau distribution

We now derive the expected equilibrium distribution under the dynamics given by Eq (4) with $F_T$ as in Eq (6). Writing the noise term more explicitly, we have

$$d(\delta\vec{x}) = [\Lambda\,\delta\vec{x} + \Gamma(\delta\vec{x}\cdot\hat{v})^3\hat{v}]dt + \sigma\,d\vec{W}_t, \tag{7}$$

where $\sigma$ is a tensor that produces the noise when applied to the standard $N_{\text{genes}}$-dimensional Wiener process $\vec{W}_t$. The time derivative of $p(\vec{x})$, the distribution of $\vec{x}$ over multiple realizations of this dynamics, is given by the Fokker–Planck equation:

$$\frac{\partial p(\vec{x})}{\partial t} = -\vec{\nabla}\cdot\left[\left(\Lambda\,\delta\vec{x} + \Gamma(\delta\vec{x}\cdot\hat{v})^3\hat{v}\right)p\right] + \frac{1}{2}\nabla^2(\sigma\sigma^T p). \tag{8}$$

We then obtain equilibrium solutions by setting $\frac{\partial p}{\partial t} = 0$. Assuming a solution of the form

$$p_L(\vec{x}) = Z^{-1}\exp\left[-\frac{1}{2}\delta\vec{x}\cdot A\cdot\delta\vec{x}^T - \frac{B}{4}(\delta\vec{x}\cdot\hat{v})^4\right], \tag{9}$$

we solve for $A$ and $B$ in terms of the dynamical parameters $\Lambda$, $\Gamma$, and $\sigma$:

$$A = -2(\sigma\sigma^T)^{-1}\Lambda \tag{10}$$

$$B = -2\left[\hat{v}^T\cdot(\sigma\sigma^T)^{-1}\cdot\hat{v}\right]\Gamma. \tag{11}$$

We call $p_L$ from Eq (9) the "Landau distribution" as we expect this shape near any continuous phase transition that can be described by simple Landau theory. In particular, Landau's theory of continuous phase transitions describes a Gibbs free energy of the form [54, 55]

$$\mathcal{F}(y) - \mathcal{F}_0 = \frac{a}{2}y^2 + \frac{b}{4}y^4. \tag{12}$$

Along the bistable dimension $\hat{v}$, the corresponding equilibrium distribution maps exactly onto the distribution $p_L(\vec{x})$ in Eq (9) that we derived from system dynamics near a transition, with $y = \delta\vec{x}\cdot\hat{v}$, $a = \hat{v}^T\cdot A\cdot\hat{v}$, $b = B$, and $p_L(y) \propto \exp - \mathcal{F}(y)$.

Note that the above derivation assumes that the bistable dimension $\hat{v}$ is also an eigenvector of the inverse noise tensor $(\sigma\sigma^T)^{-1}$. This assumption holds in our simulation tests, but would not hold for aribitrary dynamics, in which case the noise can affect the direction of the lowest-order bistability displayed by the equilibrium distribution. In this case, we expect that the equilibrium distribution near the transition will still have the form of Eq (9) (so that our fitting procedure remains unchanged), but with a modified $\hat{v}$ that is no longer simply an eigenvector of $\Lambda$, and with an effective $\Gamma$ that contains information about the shape of $F_T$ along that dimension.

## 5.5 Fitting and model selection

To fit the Landau distribution to data, we rewrite it in terms of more easily managed parameters. We first find the Gaussian approximation corresponding to PCA:

$$p_G(\vec{x}) \propto \exp\left[-\frac{1}{2}(\vec{x} - \vec{\mu})^T \cdot J \cdot (\vec{x} - \vec{\mu})\right], \tag{13}$$

with $J$ equal to the inverse of the data's covariance matrix $\Sigma$. We then express $p_L(\vec{x})$ in terms of parameters $c$ and $d$ that set the shape along the bistable dimension $\hat{v}$, which we assume to be the first principal component of the data (the dominant eigenvector of $\Sigma$):

$$p_L(\vec{x}) = Z^{-1} \exp\left[\begin{array}{c} -\frac{1}{2}(\vec{x} - \vec{\mu})^T \cdot J \cdot (\vec{x} - \vec{\mu}) \\ -\frac{c-1}{2}J_v((\vec{x} - \vec{\mu}) \cdot \hat{v})^2 \\ -\frac{d}{4}J_v^2((\vec{x} - \vec{\mu}) \cdot \hat{v})^4 \end{array}\right], \tag{14}$$

where $J_v = \hat{v}^T \cdot J \cdot \hat{v}$ and the normalization factor $Z$ is given by

$$Z = \sqrt{\frac{(2\pi)^{N-1}}{\det J}} \frac{|c|}{2d} \exp\left(\frac{c^2}{8d}\right)\mathcal{B}, \tag{15}$$

with $\mathcal{B} = K_{1/4}\left(\frac{c^2}{8d}\right)$ when $c > 0$, $\mathcal{B} = \frac{\pi}{\sqrt{2}}\left[I_{-1/4}\left(\frac{c^2}{8d}\right) + I_{1/4}\left(\frac{c^2}{8d}\right)\right]$ when $c < 0$, and $I_\alpha$ and $K_\alpha$ are the modified Bessel functions of the first and second kind, respectively. Note that $c$ and $d$ are related to $A$ and $B$ in Eq (9) (specifically $A = J + (c-1)J_v\hat{v}\hat{v}^T$ and $B = dJ_v^2$), the original Gaussian model $p_G$ corresponds to $c = 1$ and $d = 0$, and bistability corresponds to $c < 0$ and $d > 0$.

In the case in which $N_{\text{samples}} < N_{\text{genes}}$, the covariance matrix $\Sigma$ does not have full rank, and the above equations can be interpreted by remaining in the subspace with variance. This does not affect model selection because we only look for bistability along the single dimension with largest variance (the principal component).

We use the Bayesian Information Criterion (BIC) to determine whether to select the null distribution $p_G$ (Gaussian along all components) or the Landau distribution $p_L$ (bimodal along the principal component). Note that if the number of degrees of freedom for the simple Gaussian is $n_G$, the Landau distribution has $n_L = n_G + 1$ ($d$ is the extra parameter; $c$ and $\mu_v$ do not add any additional freedom because these correspond to degrees of freedom that are also in the Gaussian model). The difference in BIC values between the two models,

$$\Delta\text{BIC} = -(n_L - n_G)\log N_{\text{samples}} + 2\sum_{i=1}^{N_{\text{samples}}}\left(\log p_L(\vec{x}_i) - \log p_G(\vec{x}_i)\right), \tag{16}$$

produces our criterion for model selection (see Fig 2). We choose a threshold $\Delta\text{BIC} > 6$ to indicate strong evidence in favor of a bistable distribution [56].

The maximum likelihood mean $\mu$ will be the same as the sample mean of the data in all dimensions except for the bistable dimension, so we vary three parameters in numerically fitting the Landau distribution: $c$, $d$, and $\mu_v = \mu \cdot \hat{v}$. We constrain $d$ to have a minimum value of $10^{-3}$ to avoid numerical issues near $d = 0$.

For the two cases in which we find strong evidence for bistability, we find the following best-fit parameter values: On day 10, $c = -4.75$ and $d = 3.19$, and on day 15, $c = -6.09$ and $d = 4.95$. (Note that these values were computed with respect to data in which we took the natural logarithm of the original expression values.)

## 5.6 Interpreting results with respect to individual genes

In Fig 3, we interpret the bistable dimension $\hat{v}$ (principal component dimension) in terms of individual genes by highlighting those genes whose variance lies mostly along $\hat{v}$. We accomplish this using the measure

$$s_i = \frac{\hat{v}_i^2}{J_v \Sigma_{ii}}, \tag{17}$$

a number between 0 and 1 that indicates the proportion of gene $i$'s variance that lies along $\hat{v}$. Genes with largest $s$ are highlighted in Fig 4 and Table 1.

A related measure is the correlation coefficient $c$ between each individual gene's expression and the expression projected along $\hat{v}$. While we know, by construction, that genes more correlated with $\hat{v}$ will have more of their variance aligned with $\hat{v}$, the measure $c$ is somewhat more intuitive in the sense that individual genes with large $c$ by definition provide more information about where the system is along the bistable dimension. We demonstrate the close relationship between $s$ and $c$ in Fig 8.

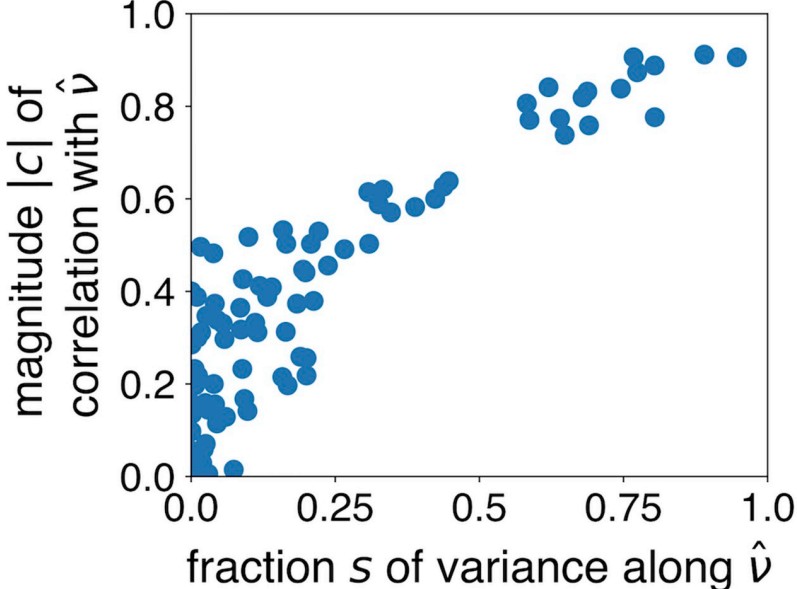

**Fig 8. Comparing two measures for the relevance of individual genes to the bistability.** The magnitude of the correlation coefficient $|c|$ between individual genes and the bistable dimension $\hat{v}$ is closely related to the proportion of variance $s$ along $\hat{v}$.

The distributions for individual gene expression shown in Fig 3 are marginals over the distribution in Eq 14:

$$
\begin{aligned}
p(x_i) &= \int p_L(\vec{x}) \prod_{j \neq i} dx_j \\
&\propto \int N(x_i, \mu_i + \eta(\hat{v} \cdot \hat{i}), \Sigma'_{ii}) \exp\left[-\frac{c}{2} J_v \eta^2 - \frac{d}{4} J_v^2 \eta^4\right] d\eta,
\end{aligned}
\tag{18}
$$

where $\Sigma'$ is the covariance matrix $\Sigma$ modified to remove variance along the bistable dimension $\hat{v}$, $N(x, \mu, \sigma^2)$ indicates a normal distribution over random variable $x$ with mean $\mu$ and variance $\sigma^2$, $J_v = \hat{v}^T \cdot J \cdot \hat{v}$, and we evaluate the final integral over $\eta$ numerically using Mathematica.

## 5.7 Simulation data and tests

We create test data using a discrete time simulation of a simple model of gene regulation:

$$
\frac{dx_i}{dt} = -x_i + \alpha \sum_j w_{ij} \tanh(x_j) + \xi,
\tag{19}
$$

here assuming the simplest all-to-all network structure for simplicity: $w_{ij} = 1$ for all $i$ and $j$. We simulate the dynamics until equilibrium using a simple Euler timestep of $\Delta t = 10^{-3}$. We ensure equilibrium by simulating to time $t_f = 100$: As in Ref. [57], we expect the maximal relaxation timescale (here with $\tau$ and $\sigma$ effectively 1 and the participation ratio $p = N_{\text{genes}}$) $t_{\max} \approx \sqrt{N_{\text{genes}}}$, so with $N_{\text{genes}} = 91$ our chosen $t_f > 10\, t_{\max}$.

In the limit of small noise, the model has a transition to bistability at $\alpha_{\text{crit}} = 1/N_{\text{genes}}$, as can be shown analytically using mean-field theory. These bifurcations can also be found numerically in non-symmetric, heterogeneous cases, in which two parameters must be tuned (as a pitchfork bifurcation has codimension 2) [58], though we do not explore this more complicated case here.

We then test our method's ability to detect the resulting bistability (Fig 1). At each value of $\alpha$, we run 100 independent trial simulations with $N_{\text{genes}} = 91$ and run the fitting analysis on the final states from each case. When the difference in BIC exceeds a threshold (here set at $\Delta \text{BIC} > 6$ [56]), we count this as a positive identification of a bistable distribution. In the case analogous to ours, with $N_{\text{samples}} = 16$ (orange points), we see that misidentifications are rare far below the transition, and identifications become easy sufficiently above the transition. As expected, with more samples ($N_{\text{samples}} = 100$, red points), it is possible to identify the incipient transition sooner, when the two modes of the distribution remain closer to one another. Note that the bifurcation point does not change with increasing $N_{\text{samples}}$, only our ability to detect the bistability. Finally, in Fig 9, we test the performance of the method in similar simulated cases with $N_{\text{genes}} = 10$ and 1000, demonstrating the method's robustness to the number of measured genes.

## 5.8 Hepatocellular carcinoma data analysis

Gene expression data from Ref. [32] were accessed via the Gene Expression Omnibus under accession number GSM155919. Data were downloaded and preprocessed using the online GEO2R tool, which applied a default log transformation. The resulting measurements corresponded to the activity of 54,675 probe sets in each of 72 samples: 10 control, 10 cirrhosis, 10 low-grade dysplastic, 7 high-grade dysplastic, 8 very early HCC, 10 early HCC, 7 advanced HCC, and 10 very advanced HCC. For each of the 7 stages of disease, we combined those

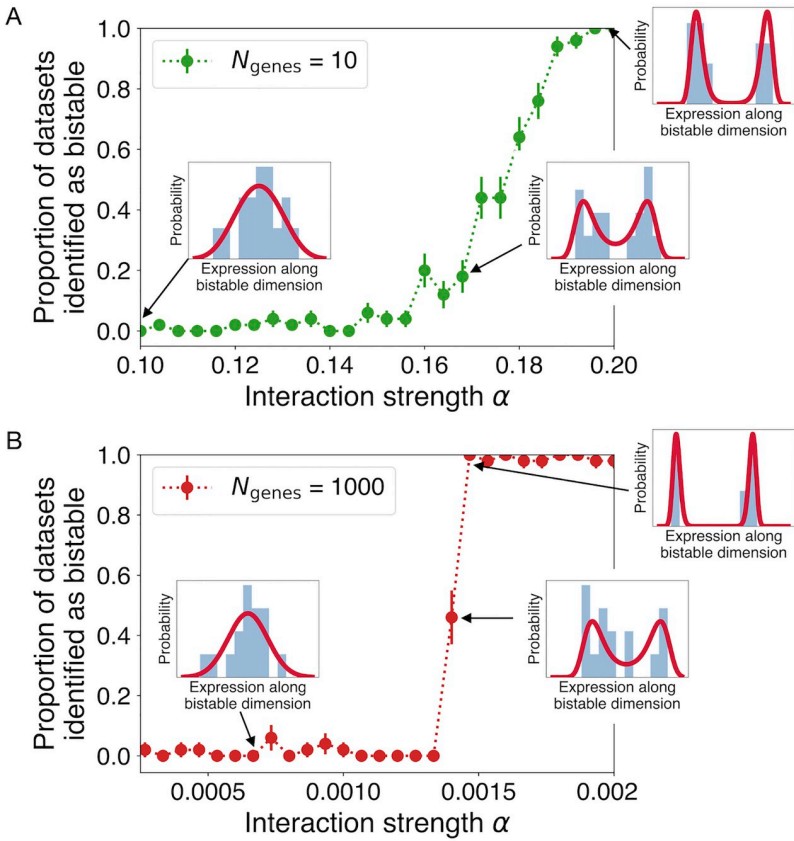

**Fig 9. The method is robust to the number of included genes.** As in Fig 1, we plot the proportion of simulated datasets in which bistability is identified as a function of interaction strength $\alpha$, here for (A) $N_{genes}$ = 10 and (B) $N_{genes}$ = 1000. Each point is a mean over 50 simulations with $N_{samples}$ = 16, with error bars indicating standard deviation of the mean.

samples with the healthy control samples to look for evidence of bimodality separating healthy from diseased expression profiles.

## Supporting information

**S1 Spreadsheet. List of genes used in this study.** Includes gene names and, for some genes, a description of possible biological functions and references to relevant literature.
(XLSX)

## Acknowledgments

We thank Cahit Ozturk for technical assistance.

## Author Contributions

**Conceptualization:** Bryan C. Daniels, Ying Wang, Robert E. Page, Jr, Gro V. Amdam.

**Data curation:** Ying Wang.

**Formal analysis:** Bryan C. Daniels.

**Funding acquisition:** Robert E. Page, Jr.

**Investigation:** Ying Wang.

**Methodology:** Bryan C. Daniels.

**Software:** Bryan C. Daniels.

**Visualization:** Bryan C. Daniels.

**Writing – original draft:** Bryan C. Daniels, Robert E. Page, Jr.

**Writing – review & editing:** Bryan C. Daniels, Robert E. Page, Jr, Gro V. Amdam.

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
