## [Decision Letter · Decision Letter 0]

30 Jan 2023

Dear Dr. Daniels,

Thank you very much for submitting your manuscript "Identifying a developmental transition in honey bees using gene expression data" for consideration at PLOS Computational Biology.

As with all papers reviewed by the journal, your manuscript was reviewed by members of the editorial board and by several independent reviewers. In light of the reviews (below this email), we would like to invite the resubmission of a significantly-revised version that takes into account the reviewers' comments.

All three reviewers ask for some clarifications in the presentation of the methods and the results. For example, Reviewer 1 recommends performing statistical tests to confirm the better performance of the presented method to the Gaussian mixture model (Figure 4, although the reviewers refers to Figure 3). Also, the choice of the 91 genes should be justified, and the robustness of the result w.r.t to this choice commented. Please also carefully consider the other comments of the three reviewers.

We cannot make any decision about publication until we have seen the revised manuscript and your response to the reviewers' comments. Your revised manuscript is also likely to be sent to reviewers for further evaluation.

Sincerely,

Carl Herrmann, Ph.D.

Academic Editor

PLOS Computational Biology

Kiran Patil

Section Editor

PLOS Computational Biology

Reviewer's Responses to Questions

**Comments to the Authors:**

Reviewer #1: Daniels et al. presents a computational method for detecting a bifurcation (of the sense of nonlinear dynamical systems) in a gene network consisting of mutually inhibiting genes (loosely speaking, see below) that governs the behavior in bees. The bifurcation is proposed to underly the developmental swicth between performing tasks in the nest and leaving the nest for foraging. The algorithm is designed to detect the appearance of a bimodal distribution of gene expression values of key regulatory genes involved in this switch in a population of bees, as a manifestation of a symmetry-breaking (pitchfork) bifurcation. The principle is essentially the fit of the distribution to a model of bimodal distribution. Specifically (the novelty), the author use a model derived from dynamical systems theory and Landau theory to compute the equilibrium state in a stochastic system in a bistable regime. They show that their model fits the data (the distribution of expression of key regulatory genes in bees of a population of N=16 bees) better than simply assuming two Gaussians (gaussian mixture model) to model the bimodal data, hence detect the bifurcation.

OVERALL:

Superficially, this work looks like a heuristic, albeit well-informed, approach of fitting distributions. But this work is more than that. It is of great originality and connects a bottom-up theory based approach (where theory encompasses dynamical systems theory and Landau theory) with the statistical approach of fitting the data to a model. And it does so in a logically coherent, hence satisfactory manner, which is rare in the current landscape of theory-free, big-data-driven bioinformatics.

While one could further dissect the logical structure of this confluence of two distinct methodological ideologies for its actual epistemological merit, which would require the skill of a historian or philosopher of science, I found, as a practitioner, this work by Daniels and coauthors most elegant and interesting, one of the most profound yet simple and enjoyable read for a long time in computational biology. While one may criticise the small number of bees analyzed, the logical and theoretical narrative is compelling, such that one could view the presented validation using real-world data as a first step, a demonstration example.

It thus appears that, modulo some simplifying assumptions in the theory and modeling that one could discuss, this work is of high scientific merit. In these days of floods of bioinformatics tools based on ad hoc heuristics, the approach presented here will likely stand out and also be most useful: the core idea is simple, but solidly rooted in theory of dynamical systems considerations, and the purpose of the method is clear from the beginning, addressing an unmet need that is well relatable by a life science audience increasingly aware of the dynamical systems principles that underlies developmental switches.

For this reason, I expect this ms. to have impact, notably in the growing field of single-cell transcriptomics. This work is also overall (modulo points below) well written, with logical clarity throughout both the theoretical as well as the biological parts.

MAJOR ISSUE:

Unfortunately, I could not see an indication of any effort to share the code.

OTHER ISSUES THAT, IF ADDRESSED MAY HELP FURTHER IMPROVE THE MANUSCRIPT:

(0) While the authors adequately present prior work in this area of detecting bifurcations, it would be useful to make a clear distinction upfront a bit more explicitly: Most prior work has been in the tradition of “Early Warning Signals” of critical transitions and seek to PREDICT an IMMINENT bifurcation (pitchfork, or saddle node) BEFORE it happens. The algorithms have a validity that is strictly limited to before the bifurcation point but not after (when bimodality is present). By contrast, the author’s new approach, similar to that of one cited reference (19), seeks to DETECT the earliest indication of bimodality AFTER the bifurcation has happened.

(1) A main question that should be presented upfront with clarity to those readers familiar with the theory of dynamical systems and concepts of bifurcations, but not the specific examples of the bee behavior, is the following: Wha interactions drives the dynamics, and what is a dynamical system defined here? This is important because the detection of bifurcation based on a statistical example of replicate instances of the very same dynamical system obliges us to define what are the interactions and what is the dynamical system whose state variables exert or are controlled by the interactions.

Instead the authors almost nonchalantly switch between three levels of interactions – and hence three levels of systems, without explicitly declaring them, which can be confusing:

(a) Interactions between genes (e.g. lines 75, or lines 167, 186 ad more): Authors talk about “gene network”. In the field of dynamical systems applied to biology it is a long tradition to use this term for INTRACELLUALR gene regulatory networks. In this case the level of the system is the CELL each with its own (identical) internal GRN, and we examine ensemble of cells

(b) Interactions between cells, tissues (e.g. lines 45 “endocrine”; line 175+ “blood levels...”): Here an INTRA-INDIVIDUAL, INTER-CELLULAR (hormonal) network is suggested. In this case the system is the organism and we examine ensemble of organism (bees)

(c) Interaction between individual bees (e.g. line 51 “interaction with other foraging bees, ... ; line 181+). It looks like there is some sort of quorum sensing in the bee colony. In this case the dynamical system is the colony.

This distinction is important for a proper formulation of the theory using the dynamical systems formalism. The method requires a stochastic ensemble of instances of systems that are in principle identical and independent (generating the distribution being measured and fitted with the model), and whose high-dimensional features (expression values of genes) that interact (and form the feedback loop) drives the dynamics; therefore one has to be clear in the system description. Strictly speaking (1) and (2) are OK, here.

But: in (3) having interactions between the systems (bees) which are the members of the distribution being evaluated, is a different matter that may complicate the model! However, as seen in other systems, the existence of such interactions may not disrupt the bifurcation behavior (but possibly bias it, e.g. making it asymmetric). But for the formalism and model it is important to have a clean description.

In any case, this classification is important for sake of being precise; the authors cannot have it multiple ways, even if I don’t think this issue will affect the strength of results. They shold be clear about the very system they examine upfront and not muddle the levels of system

(2) LANDAU model vs. GMM: In this comparison, it would be good to add two elements:

- The statistical significance for the claim, which is based only on visuals (“performs noticeably worse”) of Fig. 3, that LANDAU model is better. There are simulations for N=100 and N=50,000 in Fig 3A and B, respectively. At least for Fig 3A, error bars should be shown or described (if too small). A statistical test of significance (KS?) may be useful.

- But even if the Landau model significantly better: This per se does not mean that it is better because of the reason claimed, that it is a “bottom” up theoretical model as opposed to the just phenomenology of the distribution. Could there be other “ad hoc” non-Gaussian distribution-based mixture models that could provide a better fit of the data than the GMM? (This could be discussed or evaluated on the simulated model)

(3). FIG. 5: The comparison with the previously critical reported transition index. It is interesting that here the latter works only with genes selected a priory to be involved in the transition. Note that several groups have used this or simialr approach as a tool to actually IDENTIFY the genes involved in the transition (e.g. PMID 23230504, similar to ref 18 that the author cite ). It must be stated for clarity that (a) This transition index method is actually inherently agnostic of choice of genes. But also that (b) that in this particular case -and in others- it seems that it works best with genes that play a role in he transition. A brief discussion of this interesting observation could be provided these relevant genes may be biased to fluctuate more in the pre-transition phase already along the dimension of the bifurcation.

(4). For the simulation model (Lines 421 ff, eq 19): It is not clear how the N-dimensional system (N=91) produces a bistable dynamics. Details may be in ref. 52, but that paper is inaccessible to many. Questions to be addressed in the text: How were bistable dynamics in these simulated networks with bifurcation parameter alpha identified (search space = product of weight matrices [w_ij] and alpha values?) How exhaustive was the search, etc..)? What was criterion for bistability? (You used BIC, but between what models? – one has to guess: it is for unimodal vs bimodal distribution ?)? Why does the bifurcation point change with N_sample (earlier in larger N_sample in Fig 1)? Etc...

Also: I am not familiar with bistability in high-dimensional systems. Specifically, the pitchfork bifurcation (eq 6) is not structurally stable, at least in the ODE form, not sure how this is in the Hopfield network form. Thus, an expansion of this last section that provides more detailed explanation of this model for the simulation would be useful.

(5). Line 113: for this statement of the direction of the largest variance at bifurcation, for clarity please point to the text of lines 331+ or summarize the explanation here in main text.

Reviewer #2: The authors present a method of detecting bi-stability in gene expression. They applies their method to a honey bee development dataset.

The presentation, figures and method are nicely clear.

Nevertheless, I think this work should not be considered for publication, for the following (main) reason: it appears as an (incomplete) mixture of three different subjects, instead of a work with clearly defined contours. I suggest that the authors should focus more deeply in one of the following subjects instead mixing them together:

1) An experimental work around development transition in honey bees, with the starting point as bi-stable genes. Among these identified genes, they role should be clarified: are they transition “drivers” or consequences of the transition? What happen at the single-tissue/single cell level? What happen if these genes are knock-down/over-expressed? Such a work should include other experimental data.

2) A bioinformatic work that propose a bi-stability identification method. The author method should be applied to many other datasets that are expected to have bi-stability, eg cell differentiation datasets. Is their method able to identify genes that are known to be bistable? How does their method depend on noise in dataset? Is it necessary to have different time points? Is their method efficient for all large dimension experimental data (bulk RNA-seq, sincgle-cell RNA-seq, quantitative proteomic data, etc.)?

3) A modeling work that is able to propose a mechanistic explanation when bi-stable expression is observed. For that, they should consider the different bistable distributions (Landau, mixed gaussian with equal variance, mixed gaussian with unequal variances, etc.). They also need to consider different type of modeling framework that is able to produce a bi-stability (ODE based of chemical dynamics, ODE based on Michaelis-Menten, agent based methods, Boolean approaches, etc.).

Additional comments:

- The choice of the list of genes is not clear (first paragraph of results). Are they directly extracted from references 29/30?

- Laudau distribution “seems” to fit better than mixed gaussian one; could the authors provide a test for it? What is the exact number of free parameters for both of these distributions (it seems that mixed gaussian has 3...)?

Reviewer #3: Daniels BC et al propose a statistical method for identifying bistability near a continuous

transition directly from high-dimensional gene expression data. They applied the method

to a transcriptomic dataset of 91 pre-selected genes from honey bees at different developmental time points, showing successful identification the emergence of a bistable gene expression pattern statistically visible at age 10 days and more apparent at age 15 days.

Although model proposed is very original and of interest to a large community, further analyses are needed to evaluate the results.

Major comments

1. The proposed approach could be useful for interpreting existing gene expression data, both in the context of social insects and in other biological contexts. Authors should provide details of the computational implementation of the method and possibly make it available for downloading and testing.

2. The criteria for having fixed the data set size at 91 genes are not discussed. Simulations should be performed to test whether the number of genes analyzed could affect the method's ability to identify the transition state.

3. The authors state that the method can identify individual genes that are related to transition states through a measure of proportion of their variance s. It is not explained and shown whether the genes with the highest s are also expected to be those with the highest BIC bimodality coefficient.

4. The results and the relevance of the proposed model should be corroborated by the analysis of an independent transcriptomic dataset, such as the recently published dataset in (Zhang W et al, iScience 2022) publicly available at GEO NCBI : GSE184507 or other relevant datasets.

**Have the authors made all data and (if applicable) computational code underlying the findings in their manuscript fully available?**

Reviewer #1: **No: **data and code should be made available - I could not find indication of this, although this may still happen if accepted

Reviewer #2: Yes

Reviewer #3: **No: **

PLOS authors have the option to publish the peer review history of their article (what does this mean?). If published, this will include your full peer review and any attached files.

Reviewer #1: **Yes: **Sui Huang

Reviewer #2: No

Reviewer #3: No
---

## [Decision Letter · Decision Letter 1]

26 Jun 2023

Dear Dr. Daniels,

Thank you very much for submitting your manuscript "Identifying a developmental transition in honey bees using gene expression data" for consideration at PLOS Computational Biology. As with all papers reviewed by the journal, your manuscript was reviewed by members of the editorial board and by several independent reviewers. The reviewers appreciated the attention to an important topic. Based on the reviews, we are likely to accept this manuscript for publication, providing that you modify the manuscript according to the review recommendations.

Sincerely,

Kiran Raosaheb Patil, Ph.D.

Section Editor

PLOS Computational Biology

Kiran Patil

Section Editor

PLOS Computational Biology

Reviewer's Responses to Questions

**Comments to the Authors:**

Reviewer #1: I am satisfied with the authors' revisions - they have addressed all my specific concerns satisfactorily. Especially, of paramount importance are the three possibilities in point (1) regarding the levels of interactions, which is now discussed in sufficient detail.

A minor point: I am not a statistician to be able to say with certainty that the statement in lines 159 ff ("Comparing the best fit .." ) is correct. It sounds reasonable to me, but here the problem is that of a type II error (failing to reject the null hypothesis that the two curves being compared are the same, when it is FALSE) - The author seem to just use KS and p-values in the Type I error setting. The KS test tends to have very low p-values which sets the bar high - which is good. But is good? It should be a matter of statistical power (what should beta value be?) and presented as such instead of Type I error situation (statistical significance of difference). Just my 2 cents on this point... Some additional verbiage of clarification may help.

Reviewer #2: The author took into account my comment. They answered to all of them and modify the text accordingly.

Nevertheless, I find that the modification are quite minimal, their work still appear as an incomplete mixture:

- They applied their method to another example, but they didn't explain for what type of (omics) data their method can be used (and for which type of data their method cannot be used)

- They assumptions are "likely to hold in many biological network". This is not very precise: what are the exact mathematical definition of their assumptions? For what class of dynamics are their method applicable (and for what class of dynamics are their method not applicable).

That is why I do not include a recommendation.

Reviewer #3: The authors have responded comprehensively to all raised points, giving special attention to key aspects such as the method's robustness concerning the number of measured genes. They have also provided a more elucidating explanation of the "s" measure by drawing a comparison to the "c" correlation between the expression of individual genes and the expression projected along the

bistable dimension. Additionally, the authors have showcased the efficacy of their method on an independent transcriptomics dataset, further validating its performance and applicability.

**Have the authors made all data and (if applicable) computational code underlying the findings in their manuscript fully available?**

Reviewer #1: Yes

Reviewer #2: None

Reviewer #3: Yes

PLOS authors have the option to publish the peer review history of their article (what does this mean?). If published, this will include your full peer review and any attached files.

Reviewer #1: **Yes: **Sui Huang

Reviewer #2: No

Reviewer #3: No

Figure Files:

Data Requirements:

Reproducibility:

References:

---

## [Decision Letter · Decision Letter 2]

5 Sep 2023

Dear Dr. Daniels,

We are pleased to inform you that your manuscript 'Identifying a developmental transition in honey bees using gene expression data' has been provisionally accepted for publication in PLOS Computational Biology.

Best regards,

Mark Alber, Ph.D.

Section Editor

PLOS Computational Biology

Kiran Patil

Section Editor

PLOS Computational Biology

Reviewer's Responses to Questions

**Comments to the Authors:**

Reviewer #2: The authors updated their text according to my comments.

**Have the authors made all data and (if applicable) computational code underlying the findings in their manuscript fully available?**

Reviewer #2: Yes

PLOS authors have the option to publish the peer review history of their article (what does this mean?). If published, this will include your full peer review and any attached files.

Reviewer #2: No

---

## [Editor Report · Acceptance letter]

18 Sep 2023

PCOMPBIOL-D-22-01604R2 

Identifying a developmental transition in honey bees using gene expression data

Dear Dr Daniels,

I am pleased to inform you that your manuscript has been formally accepted for publication in PLOS Computational Biology. Your manuscript is now with our production department and you will be notified of the publication date in due course.

With kind regards,

Anita Estes
